# Machine learning of dissection photographs and surface scanning for quantitative 3D neuropathology

Harshvardhan Gazula[1], Henry FJ Tregidgo[2], Benjamin Billot[3], Yael Balbastre[1], Jonathan Williams-Ramirez[1], Rogeny Herisse[1], Lucas J Deden-Binder[1], Adria Casamitjana[2,4], Erica J Melief[5], Caitlin S Latimer[5], Mitchell D Kilgore[5], Mark Montine[5], Eleanor Robinson[2], Emily Blackburn[2], Michael S Marshall[6], Theresa R Connors[6], Derek H Oakley[6], Matthew P Frosch[6], Sean I Young[1], Koen Van Leemput[1,7], Adrian V Dalca[1,3], Bruce Fischl[1], Christine L MacDonald[8], C Dirk Keene[5], Bradley T Hyman[6], Juan E Iglesias[1,2,3]*

[1]Martinos Center for Biomedical Imaging, MGH and Harvard Medical School, Charlestown, United States; [2]Centre for Medical Image Computing, University College London, London, United Kingdom; [3]Computer Science and Artificial Intelligence Laboratory, MIT, Cambridge, United States; [4]Biomedical Imaging Group, Universitat Politècnica de Catalunya, Barcelona, Spain; [5]BioRepository and Integrated Neuropathology (BRaIN) Laboratory and Precision Neuropathology Core, UW School of Medicine, Seattle, United States; [6]Massachusetts Alzheimer Disease Research Center, MGH and Harvard Medical School, Charlestown, United States; [7]Neuroscience and Biomedical Engineering, Aalto University, Espoo, Finland; [8]Department of Neurological Surgery, UW School of Medicine, Seattle, United States

*For correspondence:
jiglesiasgonzalez@mgh.harvard.edu

**Abstract** We present open-source tools for three-dimensional (3D) analysis of photographs of dissected slices of human brains, which are routinely acquired in brain banks but seldom used for quantitative analysis. Our tools can: (1) 3D reconstruct a volume from the photographs and, optionally, a surface scan; and (2) produce a high-resolution 3D segmentation into 11 brain regions per hemisphere (22 in total), independently of the slice thickness. Our tools can be used as a substitute for ex vivo magnetic resonance imaging (MRI), which requires access to an MRI scanner, ex vivo scanning expertise, and considerable financial resources. We tested our tools on synthetic and real data from two NIH Alzheimer's Disease Research Centers. The results show that our methodology yields accurate 3D reconstructions, segmentations, and volumetric measurements that are highly correlated to those from MRI. Our method also detects expected differences between *post mortem* confirmed Alzheimer's disease cases and controls. The tools are available in our widespread neuroimaging suite 'FreeSurfer' (https://surfer.nmr.mgh.harvard.edu/fswiki/PhotoTools).

## eLife assessment

The authors of this study implemented an **important** toolset for 3D reconstruction and segmentation of dissection photographs, which could serve as an alternative for cadaveric and ex vivo MRIs. The tools were tested on synthetic and real data with **compelling** performance. This toolset could further contribute to the study of neuroimaging-neuropathological correlations.

**eLife digest** Every year, thousands of human brains are donated to science. These brains are used to study normal aging, as well as neurological diseases like Alzheimer's or Parkinson's. Donated brains usually go to 'brain banks', institutions where the brains are dissected to extract tissues relevant to different diseases. During this process, it is routine to take photographs of brain slices for archiving purposes.

Often, studies of dead brains rely on qualitative observations, such as 'the hippocampus displays some atrophy', rather than concrete 'numerical' measurements. This is because the gold standard to take three-dimensional measurements of the brain is magnetic resonance imaging (MRI), which is an expensive technique that requires high expertise – especially with dead brains. The lack of quantitative data means it is not always straightforward to study certain conditions.

To bridge this gap, Gazula et al. have developed an openly available software that can build three-dimensional reconstructions of dead brains based on photographs of brain slices. The software can also use machine learning methods to automatically extract different brain regions from the three-dimensional reconstructions and measure their size. These data can be used to take precise quantitative measurements that can be used to better describe how different conditions lead to changes in the brain, such as atrophy (reduced volume of one or more brain regions).

The researchers assessed the accuracy of the method in two ways. First, they digitally sliced MRI-scanned brains and used the software to compute the sizes of different structures based on these synthetic data, comparing the results to the known sizes. Second, they used brains for which both MRI data and dissection photographs existed and compared the measurements taken by the software to the measurements obtained with MRI images. Gazula et al. show that, as long as the photographs satisfy some basic conditions, they can provide good estimates of the sizes of many brain structures.

The tools developed by Gazula et al. are publicly available as part of FreeSurfer, a widespread neuroimaging software that can be used by any researcher working at a brain bank. This will allow brain banks to obtain accurate measurements of dead brains, allowing them to cheaply perform quantitative studies of brain structures, which could lead to new findings relating to neurodegenerative diseases.

## Introduction

Morphometric measurements such as cortical thickness and subcortical volumetry can be used as surrogate biomarkers of aging (*Salat et al., 2004*; *Walhovd et al., 2005*; *Coupé et al., 2017*) and disease (*Lerch et al., 2005*; *Desikan et al., 2009*; *Dickerson et al., 2009*; *Blanken et al., 2017*) via confirmed histopathological changes. Integrating neuroimaging tools with neuropathological assessments can enable neuropathological–neuroimaging correlations for studying neurodegenerative diseases, that is, connecting macroscopic imaging with histological ground truth derived from microscopic imaging (*Webster et al., 2021*). While *ante mortem* magnetic resonance imaging (MRI) studies provide accurate and reliable morphometric data, they are often unavailable or occur too long before death, impeding reliable histopathological correlation. Cadaveric MRI can circumvent these challenges, but logistic and legal issues often complicate this procedure.

An alternative to cadaveric imaging is ex vivo MRI, which enables high-resolution image acquisition free of subject motion and other physiological noise (*Edlow et al., 2019*). However, ex vivo MRI also has disadvantages, including: tissue degradation due to bacteria and autolysis from death to initiation of tissue fixation; cross-linking of proteins due to fixation that dramatically changes MRI properties of the tissue; and image artifacts caused by magnetic susceptibility interfaces that do not occur in vivo (*Shatil et al., 2016*).

While ex vivo MRI is relatively uncommon in brain banks that seek to establish neuropathological–neuroimaging correlations (*Ravid, 2009*; *Love, 2005*), dissection photography is routine in nearly every brain bank. Collected specimens are typically dissected into coronal slices and photographed before further blocking and histological analysis. These photographs, often underutilized, are an invaluable information resource that, if leveraged appropriately, can play a vital role in advancing our understanding of various brain functions and disorders — mainly when ex vivo MRI is unavailable. To this end, we propose a novel software suite that, for the first time, enables three-dimensional (3D)

reconstruction and quantitative 3D morphometry of dissection photographs, powered by modern machine learning techniques (*Goodfellow et al., 2016*) and 3D surface scanning (*Salvi et al., 2004*). Provided that the photographs satisfy some basic requirements (presence of a ruler or fiducial markers to estimate pixel sizes), our tools enable volumetric analysis of brain structures from the photographs, computationally guided dissection via automated segmentation, and accurate spatial mapping for neuropathological–neuroimaging correlation.

Our suite is freely available and includes three modules. The first module is a set of preprocessing routines for the photographs that enables the correction of perspective and calibration of pixel sizes. The second module is a joint image registration algorithm that allows 3D reconstruction of the photographs using a 3D surface scan of the brain as a reference. This module can also use a probabilistic atlas as a reference for the reconstruction, thus circumventing the need for a surface scanner. This scan-free mode enables retrospective analysis of photographs without corresponding 3D scans, albeit with lower accuracy than if surface scanning was available. The third and final modules use machine learning to provide a high-resolution 3D image segmentation of the reconstructed stack. This module combines a state-of-the-art deep segmentation neural network (a U-Net, *Ronneberger et al., 2015*) with a domain randomization approach (*Billot et al., 2023a*), thus enabling analysis of photographs with different intensity profiles, for example, acquired under various illumination conditions, with different cameras or camera settings, or from fixed or fresh tissue. Moreover, the machine learning method enables the estimation of 'smooth', isotropic segmentations that accurately interpolate across the gaps between the coronal planes captured in the photographs.

We note that this article extends our previous conference paper (*Tregidgo et al., 2020*) by (1) improving upon the 3D reconstruction methods; (2) using machine learning (rather than Bayesian methods) to produce segmentations that are more accurate and also isotropic (i.e., provide labels in between slices); and (3) providing extensive experiments on different synthetic and real datasets. The rest of this article is organized as follows. First, we present results on three different datasets, both

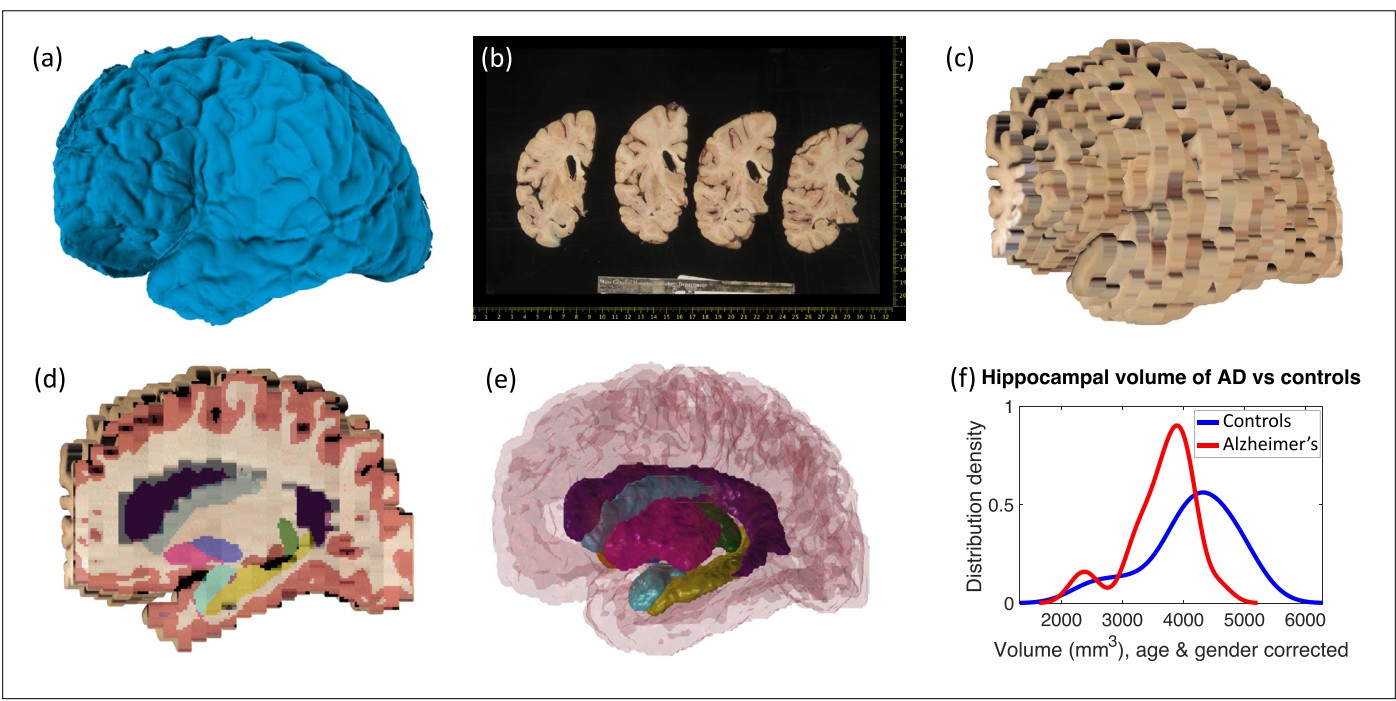

**Figure 1.** Examples of inputs and outputs from the MADRC dataset. (**a**) Three-dimensional (3D) surface scan of left human hemisphere, acquired prior to dissection. (**b**) Routine dissection photography of coronal slabs, after pixel calibration, with digital rulers overlaid. (**c**) 3D reconstruction of the photographs into an imaging volume. (**d**) Sagittal cross-section of the volume in (**c**) with the machine learning segmentation overlaid. The color code follows the FreeSurfer convention. Also, note that the input has low, anisotropic resolution due to the large thickness of the slices (i.e., rectangular pixels in sagittal view), whereas the 3D segmentation has high, isotropic resolution (squared pixels in any view). (**e**) 3D rendering of the 3D segmentation into the different brain regions, including hippocampus (yellow), amygdala (light blue), thalamus (green), putamen (pink), caudate (darker blue), lateral ventricle (purple), white matter (white, transparent), and cortex (red, transparent). (**f**) Distribution of hippocampal volumes in *post mortem* confirmed Alzheimer's disease vs controls in the MADRC dataset, corrected for age and gender.

**Table 1.** Area under the receiver operating characteristic curve (AUROC) and p-value of a non-parametric Wilcoxon rank sum test comparing the volumes of brain regions for Alzheimer's cases vs controls.

The volumes were corrected by age and sex using a general linear model. We note that the AUROC is bounded between 0 and 1 (0.5 is chance) and is the non-parametric equivalent of the effect size (higher AUROC corresponds to larger differences). The sample size is $N = 33$.

| Region | Wh matter | Cortex | Vent | Thal | Caud | Putamen | Pallidum | Hippoc | Amyg |
|---------|-----------|--------|------|------|------|---------|----------|--------|------|
| AUROC | 0.45 | 0.52 | 0.73 | 0.48 | 0.65 | 0.64 | 0.77 | 0.75 | 0.77 |
| p-value | 0.666 | 0.418 | 0.016 | 0.596 | 0.086 | 0.092 | 0.005 | 0.009 | 0.007 |

synthetic (which enable fine-grained analysis with known ground truth) and real (which enables evaluation in real-world scenarios). Next, we discuss these results and their impact on quantitative *post mortem* neuroimaging without MRI. Finally, the Methods section elaborates on the technicalities of the preprocessing steps, the reconstruction algorithm, and the deep learning segmentation method.

## Results

### Volumetric group study of *post mortem* confirmed Alzheimer's disease

One of the main use cases of our tools is the volumetric analysis of different cerebral regions of interest (ROIs) from dissection photographs, without requiring cadaveric or ex vivo MRI. We used our tools to analyze 21 *post mortem* confirmed Alzheimer's disease (AD) cases from the Massachusetts Alzheimer's Disease Research Center (MADRC), as well as 12 age-matched controls ($N$ = 33 in total). We note that these cases comprise thick (~10 mm) slabs, sliced by hand, without cutting guides – as cutting on anatomic landmarks was prioritized over consistent slice thickness. Therefore, this dataset is representative of a challenging, real-world scenario.

Examples of the inputs and outputs of the pipeline can be found in *Figure 1* and *Video 1* in the supplementary material (also available at https://youtu.be/wo5meYRaGUY). The 3D surface scan (*Figure 1a*) reveals the coarse shape of the specimen, in this case a left hemisphere. Our preprocessing tools correct for the perspective and pixel size of routine dissection photographs (b). Then, the 3D reconstruction tool uses information from the surface scan to produce a 3D reconstruction of the photographs into an imaging volume (c). This volume consists of highly anisotropic voxels, since the slice thickness is much larger than the pixel size of the photograph. The 3D reconstruction is fed to our machine learning segmentation method (Photo-SynthSeg), which produces a high-resolution, isotropic segmentation (d, e), independently of the slice thickness. The segmentations are then used to compare the volumes of brain ROIs between the two groups (e.g., the hippocampus, as in *Figure 1f*).

*Table 1* shows the area under the receiver operating characteristic curve (AUROC) and the p-value for non-parametric Wilcoxon rank sum tests (*Mann and Whitney, 1947*) comparing the ROI volumes of AD vs controls; we leave the accumbens area and ventral diencephalon out of the analysis as their segmentations are not reliable due to poor contrast (*Fischl et al., 2002*). We note that the AUROC is the non-parametric equivalent of the effect size; a value of 0.5 represents chance, while 1.0 represents perfect separation. Age and gender were corrected with a general linear model, whereas volumes of contralateral ROIs were averaged when full brains (rather than hemispheres) were available. Our method successfully captures well-known atrophy patterns of AD, such as hippocampal atrophy and ventricle enlargement.

### Quantitative evaluation of segmentation with Photo-SynthSeg

While the AD experiment illustrates the ability of our method to detect differences in real-world data, it is crucial to specifically assess the

Machine learning of dissection photographs and surface scanning for quantitative 3D neuropathology

H Gazula, HFJ Tregidgo, B Billot, ... Williams-Ramirez, R Herisse, A Casamitjana, EJ Melief, CS Latimer, ... more, M Montine, ED Robinson, E Blackburn, MS Marshall, TR Connor, ... Oakley, MP Frosch, K Van Leemput, AV Dalca, B Fischl, CL Mac Donald, CD Keene, BT Hyman, and JE Iglesias

*Supplementary video: overview of the proposed method*

**Video 1.** Overview of the proposed method.
https://elifesciences.org/articles/91398/figures#video1

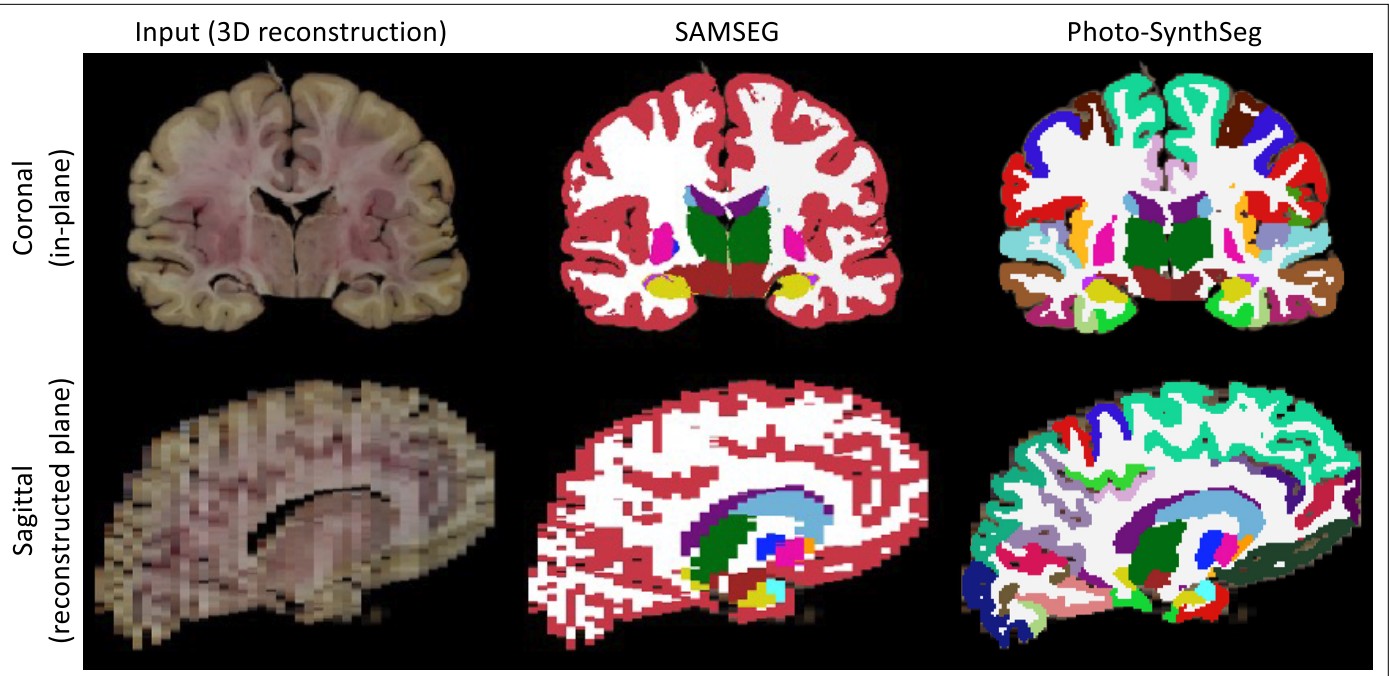

**Figure 2.** Qualitative comparison of SAMSEG vs Photo-SynthSeg: coronal (top) and sagittal (bottom) views of the reconstruction and automated segmentation of a sample whole brain from the UW-ADRC dataset. Note that Photo-SynthSeg supports subdivision of the cortex with tools of the SynthSeg pipeline.

accuracy of our segmentation and 3D reconstruction methods. To evaluate Photo-SynthSeg, we 3D reconstructed the brain volumes from the photographs of 24 cases from the AD Research Center at the University of Washington (UW-ADRC), which were cut into uniform 4 mm thick slices. Crucially, isotropic FLAIR MRI scans were acquired for these specimens ex vivo prior to dissection, which enables the use of MRI as gold standard.

We compare Photo-SynthSeg against 'SAMSEG', which is (to the best of our knowledge) the only available competing method. SAMSEG is a segmentation algorithm which we originally conceived for brain MRI (*Puonti et al., 2016*), and which employs Bayesian techniques to segment MRI scans irrespective of their contrast and pulse sequence. As we showed in *Tregidgo et al., 2020*, this technique can be adapted to 3D reconstructed photographs. *Figure 2* shows the segmentation of an UW-ADRC case using both methods. As opposed to SAMSEG, Photo-SynthSeg effectively 'interpolates' the segmentation in between slices, independently of their thickness, and is more robust against uneven intensities than the Bayesian method. Photo-SynthSeg also includes a volumetric parcellation of the cortex based on the original SynthSeg pipeline (*Billot et al., 2023a*), which relies on the Desikan–Killiany atlas (*Desikan et al., 2006*).

To evaluate Photo-SynthSeg and SAMSEG quantitatively, we computed Dice scores against manual segmentations made on a single selected slice per subject. This slice is visually chosen to be close to the mid-coronal plane, while maximizing visibility of subcortical structures; an example of such slice, along with the manual and automated segmentations, is shown in *Appendix 1—figure 3*. The Dice scores are displayed in *Figure 3*; as in the previous analysis, the accumbens area and ventral diencephalon are left out of the analysis. The figure also shows two ablations: using a probabilistic atlas instead of the case-specific reference; and using a version of Photo-SynthSeg dedicated to 4 mm slice thickness (i.e., the thickness of the UW-ADRC dataset, rather than using the general, thickness-agnostic model). The former assesses the impact of having to rely on a generic atlas when surface scans are not available, whereas the latter evaluates the ability of our neural network to adapt to thicknesses that are not known a priori (thanks to our domain randomization approach). The results show that Photo-SynthSeg generally outperforms SAMSEG, producing Dice scores over 0.8 for all structures except the amygdala – even though SAMSEG is better at segmenting the ventricles, thanks to their large size and strong contrast. The plots also show that using the probabilistic atlas produces realistic enough

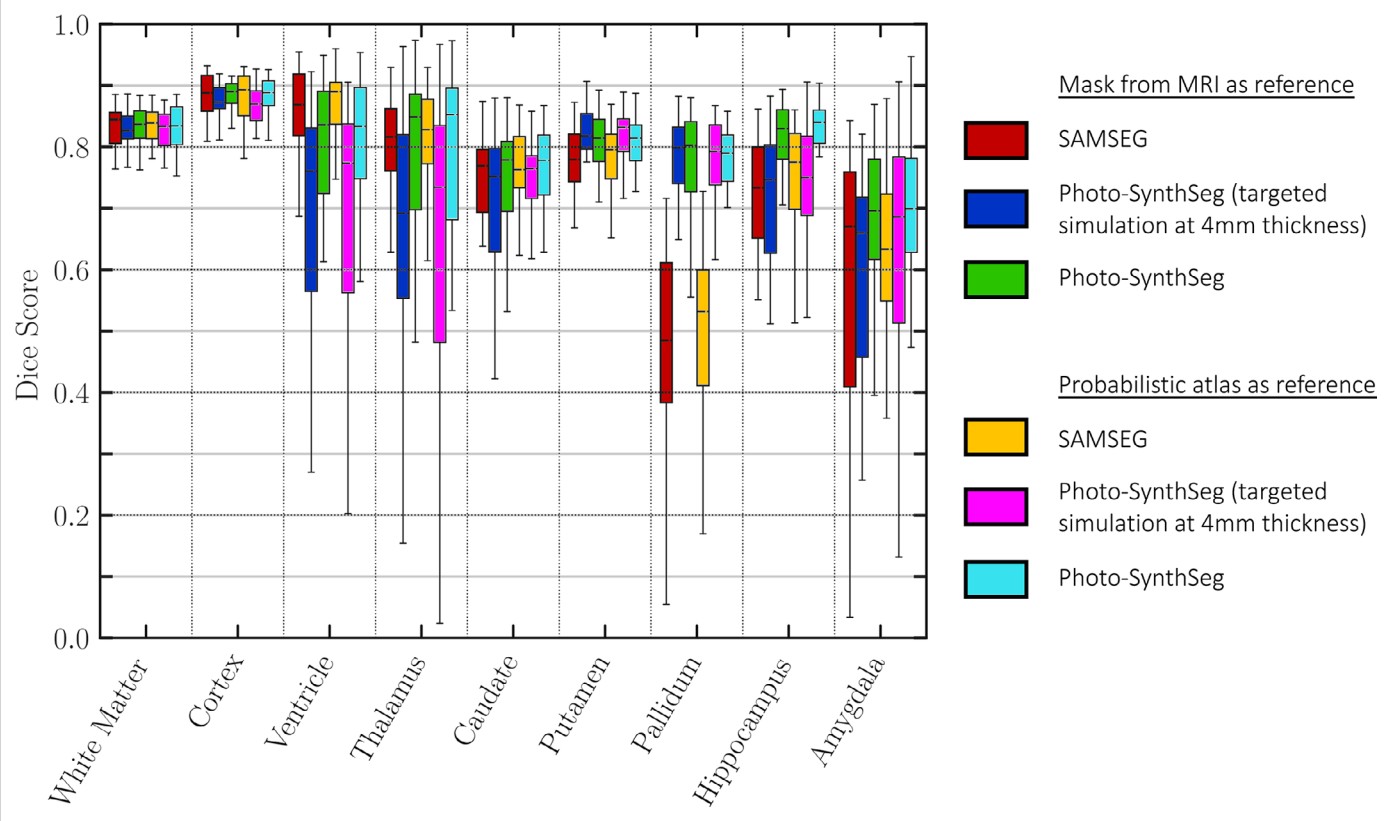

**Figure 3.** Dice scores of automated vs manual segmentations on select slices. Box plots are shown for SAMSEG, Photo-SynthSeg, and two ablations: use of probabilistic atlas and targeted simulation with 4 mm slice spacing. Dice is computed in two-dimensional (2D), using manual segmentations on select slices. We also note that the absence of extracerebral tissue in the images contributes to high Dice for the cortex.

reconstructions, such that the two-dimensional (2D) Dice scores remain high. Finally, the results also show the superiority of the thickness randomization strategy over the fixed spacing strategy, even if the latter had access to the ground truth spacing of the photographs. Even though the reader may initially find this result counterintuitive, it is consistent with our previous findings in MRI segmentation, where domain randomization strategies also outperformed targeted simulations (*Billot et al., 2020*).

This evaluation with Dice scores above is direct, but: (1) is based on a relatively small number of slices, and (2) disregards the ultimate purpose of segmentation, which is downstream analysis in 3D (e.g., volumetry). For this purpose, we also indirectly evaluated the methods by analyzing the volumes of brain ROIs derived from the segmentations of the whole stack. Specifically, we correlated these volumes with silver standard values derived from the isotropic FLAIR MRI scans using our 'standard' SynthSeg for MRI (*Billot et al., 2023a*). *Table 2* shows the correlations and p-values for Steiger tests comparing the correlation coefficients achieved by SAMSEG and Photo-SynthSeg – while considering their dependency due to the common ground truth sample. The results show once more that Photo-SynthSeg outperforms SAMSEG for nearly every structure – and in the few cases in which it does not, the Steiger test does not yield statistically significant differences. We note that the correlations are above 0.8 for most brain structures, indicating that the 3D reconstructed photographs yield usable volumes in volumetric analysis. We also note that the correlations are slightly but consistently lower for the reconstructions with the probabilistic atlas, yielding correlations close to 0.8 for most brain regions.

## Quantitative evaluation of reconstruction with digitally sliced MRI data

In addition to the segmentation, it is also desirable to evaluate the 3D reconstruction algorithm with the registration error (in mm). Measuring such error with real data would require manual annotation of pairs of matching landmarks, which is labor-intensive, error-prone, and not reproducible. Instead, we use simulated (digitally sliced) data created from MRI scans. While errors estimated this way may be

**Table 2.** Correlations of volumes of brains regions estimated by SAMSEG and Photo-SynthSeg from the photographs against the ground truth values derived from the magnetic resonance imaging (MRI).
The p-values are for Steiger tests comparing the correlations achieved by the two methods (accounting for the common sample).

| | Mask from MRI as reference | | | Probabilistic atlas as reference | | |
| --- | --- | --- | --- | --- | --- | --- |
| | SAMSEG | Photo-SynthSeg | p-value | SAMSEG | Photo-SynthSeg | p-value |
| White matter | 0.935 | 0.981 | 0.0011 | 0.886 | 0.935 | 0.0117 |
| Cortex | 0.930 | 0.979 | 0.0001 | 0.889 | 0.920 | 0.0366 |
| Ventricle | 0.968 | 0.988 | 0.0004 | 0.980 | 0.993 | 0.0006 |
| Thalamus | 0.812 | 0.824 | 0.4350 | 0.812 | 0.824 | 0.4252 |
| Caudate | 0.719 | 0.779 | 0.2525 | 0.733 | 0.792 | 0.2062 |
| Putamen | 0.904 | 0.779 | 0.9923 | 0.872 | 0.792 | 0.9598 |
| Pallidum | 0.727 | 0.694 | 0.6171 | 0.676 | 0.658 | 0.5698 |
| Hippocampus | 0.830 | 0.757 | 0.8873 | 0.764 | 0.776 | 0.4293 |
| Amygdala | 0.598 | 0.703 | 0.1663 | 0.576 | 0.763 | 0.0221 |

optimistic compared with real sliced tissue, this approach enables us to analyze the error as a continuous function of slice thickness without manual annotation effort.

For this purpose, we used 500 subjects from the Human Connectome Project (HCP) dataset, which includes T1- and T2-weighted MRI scans acquired at 0.7 mm isotropic resolution. After skull stripping with FreeSurfer (*Fischl, 2012*), we simulated dissection photographs and matching surface scans by: (1) digitally slicing the T2 scans and (2) using the surface of the T1 as a 3D reference to reconstruct the T2 slices. Specifically, we simulated slices with $S \times 0.7$ mm thickness ($S = 2, 4, 8, 16$), including random affine transforms and illumination fields for every simulated slice (see example in *Appendix 1—figure 1*). While we could use a nonlinear model, the results would depend heavily on the strength of the simulated deformation. Instead, we keep the warps linear as we believe that the value of this experiment lies in the trends that the errors reflect, that is, their relative rather than absolute value.

After digitally distorting the images, we used our method to 3D reconstruct the slices into their original shape. The registration error was calculated as the mean voxel displacement in mm between the reconstructed and ground truth T2 slices. Additionally, to test the robustness of the reconstruction algorithm to uneven slice spacing during dissection, we also analyze the error when a random variation of the nominal thickness (thickness jitter) is introduced for every slice.

*Figure 4* shows the box plot for the mean reconstruction error as a function of the slice spacing and thickness jitter. The results show that the reconstruction error is reasonably robust to increased slice spacing. On the other hand, thickness jitter yields greater increases in reconstruction error, particularly at larger slice spacings. This result highlights the importance of keeping the slice thickness as constant as possible during acquisition.

## Discussion

Neuroimaging to neuropathology correlation studies explore the relationship between gold-standard pathological diagnoses and imaging phenotypes by transferring the microscopic signatures of pathology to in vivo MRI. A significant impediment to this effort is the lack of quantitative tools for *post mortem* tissue analysis. For example, quantitative measurements such as cortical thickness and specific regional atrophy are often estimated qualitatively from 2D coronal slices during gross examination. A solution to this problem is leveraging dissection photography of these coronal slices routinely acquired before histology. By designing algorithms to reconstruct 3D volumes from 2D photographs and subsequently segment them, we have enabled a cost-effective and time-saving link between morphometric phenotypes and neuropathological diagnosis.

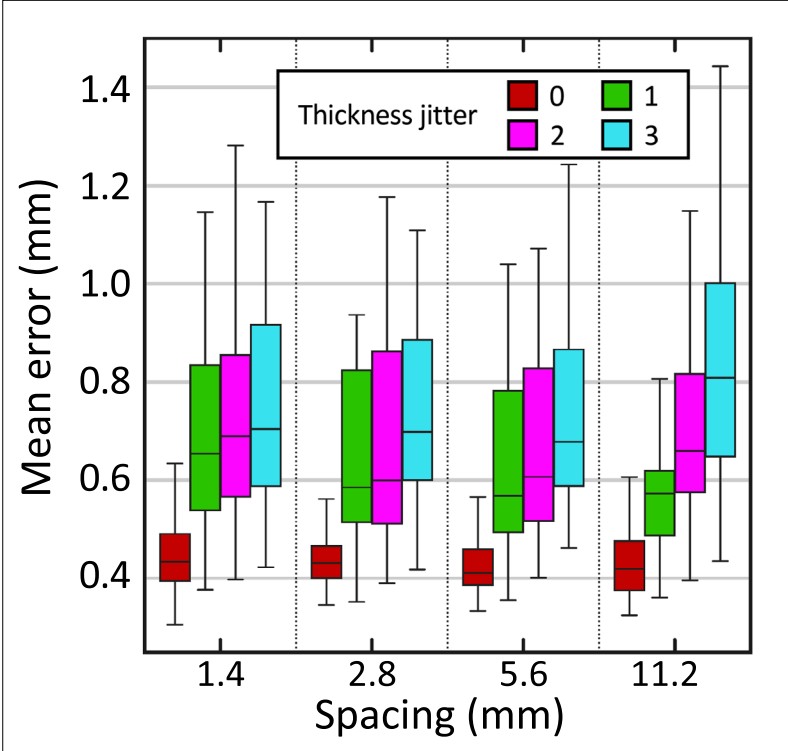

**Figure 4.** Reconstruction error (in mm) in synthetically sliced HCP data. The figure shows box plots for the mean reconstruction error as a function of spacing and thickness jitter. A jitter of $j$ means that the $n^{th}$ slice is randomly extracted from the interval $[n - j, n + j]$ (rather than exactly $n$). The center of each box represents the median; the edges of the box represent the first and third quartiles; and the whiskers extend to the most extreme data points not considered outliers (not shown, in order not to clutter the plot).

Our new tools are publicly available and utilize modern deep learning techniques and surface scanning. Being available as part of FreeSurfer makes the tools easy to use by anyone with little or no training. Furthermore, the absence of tunable parameters in Photo-SynthSeg makes the results produced by our methods highly reproducible, and the robustness of the tools enables widespread application to heterogeneous datasets. This robustness has been demonstrated by applying the tools to images from two different biobanks, with different slice thickness, tissue processing, and photographic setup. On the UW-ADRC dataset, for which MRI scans were available, we achieved correlations above 0.8 between the volumes derived from the photographs and the ground truth obtained from the MRI.

Retrospective reconstruction of datasets with no accompanying surface scan available can be achieved with a probabilistic atlas. However, such a reconstruction is laden with ambiguities because the probabilistic atlas does not have access to the true shape of the tissue – which the surface scan directly measures. Whether the increased reconstruction error is tolerable depends on the downstream task, for example, shape analysis vs volumetry.

While deep learning segmentation tools are increasingly common in medical imaging research, their practical applicability in modalities with highly varying appearance (like dissection photography) has been hindered by their limited generalization ability. Photo-SynthSeg circumvents this problem by building on our recent work on domain randomization (*Billot et al., 2023a*), and can segment 3D reconstructed stacks of photographs irrespective of the thickness of the slices and of the contrast properties of the tissue. Compared with SAMSEG, Photo-Synthseg also has the advantage of estimating the segmentation in between slices (*Figure 2*). Moreover, Photo-Synthseg inherits the computational efficiency of neural networks, and can segment a whole case in a few seconds (or tens of seconds if no graphics processing unit [GPU] is available) without any special requirements on machine learning expertise – thus enabling broad applicability.

While registration and volumetric segmentation enable morphometry and neuropathology–neuroimaging correlation, precise white matter and pial surface placement on 3D reconstructed photographs are crucial for accurate cortical analyses – for example, producing topologically correct segmentations, as opposed to volumetric segmentations that can, for example, leak across gyri. In the future, we will extend Photo-SynthSeg to enable surface analysis for cortical placement and parcellation. While the convoluted nature of cortical surfaces makes this task difficult for photographic volumes, integrating the triangular mesh provided by the surface scanner could enable accurate surface placement.

Another direction of future work will be extending the tools to axial and sagittal slices of the cerebellum and brainstem. While adapting the 3D reconstruction (*Equation 1*) is straightforward, the U-Net will need additional image synthesis and manual labeling efforts – particularly if one wishes to include new regions, such as brainstem nuclei. Additional future analyses will include: correlating the segmentation-derived volumes with clinical scores, disease subtypes, and disease duration; using techniques like SynthSR (*Iglesias et al., 2023*) to improve the resolution of the reconstructed volumes; exploring nonlinear deformation models for the 3D reconstruction; fully automatizing tissue segmentation from the background using neural networks; and extending the tools to 3D analysis of histological sections.

Leveraging the vast amounts of dissection photographs available at brain banks worldwide to perform morphometry is a promising avenue for enhancing our understanding of various neurodegenerative diseases. Our new tools will allow extraction of quantitative phenotypical information from these photographs – and thus augmentation of histopathological analysis. We expect this new methodology to play a crucial role in the discovery of new imaging markers to study neurodegenerative diseases.

# Materials and methods
## Datasets
### MADRC
Dissection photography of fixed tissue and companion surface scans for 76 cases from the Massachusetts Alzheimer's Research Center (18 whole cerebrums and 58 hemispheres). Ruling out cases with frontotemporal dementia and other comorbidities, as well as subjects that did not pass manual quality control (by JWR, RH, and LJD), led to a final sample size of $N = 33$ (21 *post mortem* confirmed Alzheimer's and 12 controls). The surface scans were acquired using a turntable with an Einscan Pro HD scanner (Shining 3D, Hangzhou, China, 0.05 mm point accuracy). Slices with variable thickness were cut with a dissecting knife on a predefined set of landmarks and photographed with a 15.1 MP Canon EOS 50D Digital SLR camera. Further details on the dissection and processing of the specimens can be found in Appendix 1.

### UW-ADRC
Dissection photography of fixed tissue and companion ex vivo MRI scans for 24 cases (all of them with both hemispheres) from the Alzheimer's Disease Research Center at the University of Washington. The MRI scans were acquired at 0.8 mm isotropic resolution using a FLAIR sequence. Coronal slices were cut with 4 mm thickness using a modified deli slicer and photographed with a 35 Megapixel (MP) camera. While no 3D surface scanning was available for this dataset, we obtained surfaces by skull stripping the MRI scans and meshing the brain surface. This dataset enables us to compute volumetric measurements from the 3D reconstructed photographs and compare them with reference values obtained from the corresponding MRI scans. Furthermore, two experienced labelers manually traced the contour of nine brain regions (white matter, cortex, lateral ventricle, thalamus, caudate, putamen, pallidum, hippocampus, and amygdala) in one slice per case, which enables computation of 2D Dice scores. Further details on the dissection and processing of the specimens can be found in Appendix 1 and *Latimer et al., 2023*.

### HCP
T1- and T2-weighted MRI scans of 500 subjects from the Human Connectome Project, acquired at 0.7 mm isotropic resolution. The scans were skull stripped using FreeSurfer (*Fischl, 2012*). We use these scans to simulate dissection photographs and matching surface scans by digitally slicing the

T2 scans and using the T1 as a 3D reference to reconstruct the T2 slices. Further details on the MRI acquisition can be found in *Van Essen et al., 2012*.

### T1-39

39 T1-weighted MRI scans at 1 mm isotropic resolution, with manual volumetric segmentations for 36 brain structures, which we used to train Photo-SynthSeg. We note that this is the labeled dataset that was used to build the probabilistic atlas in FreeSurfer (*Fischl et al., 2002*). The 36 structures include 22 that are segmented by Photo-Synthseg: left and right white matter, cortex, ventricle, thalamus, caudate, putamen, pallidum, hippocampus, amygdala, accumbens area, and ventral diencephalon. The remaining 14 labels include: four labels for the cerebellum (left and right cortex and white matter); the brainstem; five labels for cerebrospinal fluid regions that we do not consider; the left and right choroid plexus; and two labels for white matter hypointensities in the left and right hemispheres.

## Surface scanning

Surface scanning (also known as 'profilometry') is a technology that is becoming increasingly inexpensive (a few thousand dollars), mainly via structured light technology (*Salvi et al., 2004*). The preferred version of our proposed pipeline relies on a surface scan of the specimen (*Figure 1a*) acquired before slicing. This surface scan, represented by a triangular mesh, is used as an external reference to guide the 3D reconstruction of the photographs (details below). While there are no specific technical requirements for the surface scan (e.g., minimum resolution), minimizing geometric distortion (i.e., deformation) between scanning and subsequent slicing is crucial. The surface scan may be acquired with a

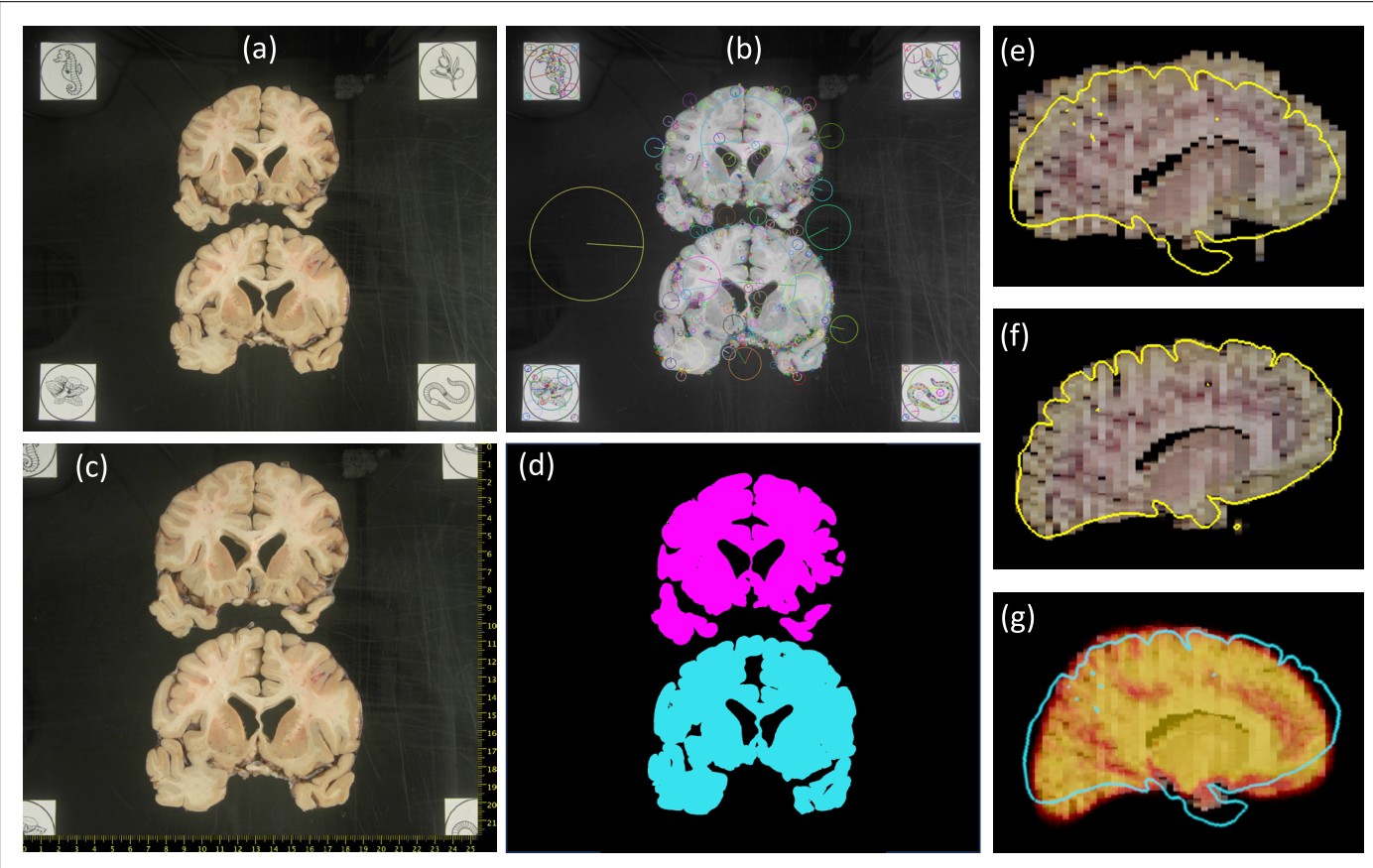

**Figure 5.** Steps of proposed processing pipeline. (**a**) Dissection photograph with brain slices on black board with fiducials. (**b**) Scale-invariant feature transform (SIFT) features for fiducial detection. (**c**) Photograph from (**a**) corrected for pixel size and perspective, with digital ruler overlaid. (**d**) Segmentation against the background, grouping pieces of tissue from the same slice. (**e**) Sagittal slice of the initialization of a three-dimensional (3D) reconstruction. (**f**) Corresponding slice of the final 3D reconstruction, obtained with a surface as reference (overlaid in yellow). (**g**) Corresponding slice of the 3D reconstruction provided by a probabilistic atlas (overlaid as a heat map); the real surface is overlaid in light blue for comparison.

handheld scanner with the specimen placed on a table, or with the scanner on a tripod and the sample on a turntable. As mentioned earlier, our pipeline does not strictly require the surface scan, as it can be replaced with a probabilistic atlas with a slight loss of accuracy (as shown in the Results section).

## Tissue slicing and photography

In preparation for the downstream 3D reconstruction, some requirements exist for the dissection and photography of the brain slabs. Specifically, our pipeline assumes that approximately parallel slices are cut in coronal direction – which is the standard practice in most brain banks. We further assume that the thickness of these slices is approximately constant, as 'thickness jitter' has a detrimental effect on the results (*Figure 4*).

Moreover, we require the presence of fiducials to enable pixel size calibration and perspective correction. Ideally, four fiducials are placed on the corners of a rectangle of known dimensions (*Figure 5a*). In the absence of such fiducials, we require the presence of at least a ruler or two orthogonal rulers; the former enables pixel size calibration via image scaling, whereas the latter enables approximate perspective correction by fitting an affine transform.

Our pipeline allows for multiple slices to be present in one photograph but requires that all photographs are of the same side of the slices (either anterior or posterior) and that the inferior–superior direction of the anatomy is approximately aligned with the vertical axis of the image (as in *Figure 5a*). Ideally, the slices should be photographed on a flat board with a solid background color that stands out from the brain tissue, as stronger contrast between tissue and background greatly facilitates image segmentation when preprocessing the photographs (further details below).

## Preprocessing of photographs

Image preprocessing starts with geometric correction, that is, pixel size calibration and perspective correction. In the ideal scenario with four fiducials, the widespread widespread scale-invariant feature transform (SIFT, *Lowe, 1999*) is used to detect such fiducials (*Figure 5b*) and compute a spatial transform that corrects for perspective distortion and calibrates the pixel size – see *Figure 5c*, where the pixel size calibration enables superimposition of a digital ruler.

Our software also supports a manual mode where the user clicks on two, three, or four landmarks: two points with a known distance in between (enables approximate pixel size calibration); three points at the ends of two rulers plus their intersection (enables approximate perspective correction with an affine transform); or four points on the corners of a rectangle of known dimensions (enables full perspective correction). This manual mode is useful when no fiducials are present, but the user can still identify features with known dimensions in the photograph, for example, on an imprinted grid pattern, or along rulers.

After geometric correction, our methods require a binary segmentation of the brain tissue, separating it from the background. While our tools do not have any specific requisites in terms of background, using a solid background with a distinct color (e.g., a 'green screen') can greatly facilitate segmentation; otherwise, more extensive manual intervention may be needed. In our experiments, using a flat black background enabled us to automatically segment the tissue with a combination of thresholding and morphological operations, keeping manual edits to a minimum (a couple of minutes per photograph, mostly to erase bits of cortical surface that are sometimes visible around the edge of the face of the slice).

Given this binary mask, the final preprocessing step requires the user to specify the order of the slices within the photograph – which may be anterior to posterior or vice versa, but must be consistent within and across photographs of the same case. Moreover, the user also needs to specify whether two connected components belong to the same slice, which often happens around the temporal pole. This can be quickly accomplished with a dedicated graphical user interface that we distribute with our tools (*Figure 5d*).

## 3D volumetric reconstruction from photographs

Recovering a 3D volume from a stack of 2D images entails a consistent 3D reconstruction of the stack via joint image alignment – known as 'registration' (*Maintz and Viergever, 1998*; *Pluim et al., 2003*; *Zitová and Flusser, 2003*; *Sotiras et al., 2013*). We pose 3D reconstruction as a joint optimization problem (*Pichat et al., 2018*; *Mancini et al., 2019*) and use a 3D reference volume for the

registration. This volume is ideally a binary 3D mask obtained by rasterizing (filling) the triangular mesh from the surface scan. If a surface scan is not available, one can instead use a probabilistic atlas (*Shattuck et al., 2008*) of brain shape, which provides a rough reference for the reconstruction. However, this reference cannot drive the reconstruction toward the actual shape of the specimen nor correct deviations from the user-provided slice thickness.

Given $N$ photographed slices and their corresponding masks, the goal of this optimization framework is to simultaneously identify: (1) a set of $N$ 2D affine geometric transforms $\{\phi_n\}_{n=1,\dots,N}$ (rotation, translation, shear, and scaling for each slice); (2) a scaling factor $s$ in the anterior–posterior direction shared by all slices; and (3) a rigid 3D transform $\Psi$ for the reference (rotation and translation). These transforms seek to align the slices with each other and with the reference volume.

We note that affine transforms (rather than rigid) are required for the photographs due to imperfections in the image preprocessing of the previous section; nevertheless, we expect the shear and scaling of these transforms to be small. Furthermore, we use affine rather than nonlinear transforms because the latter compromise the robustness of the registration, as they introduce huge ambiguity in the space of solutions (e.g., one could add an identical small nonlinear deformation to every slice almost without changing the feature of the 3D reconstruction). This affine model makes it particularly important to place connected components of the same slice in a correct relative position when there is more than one, for example, in the temporal pole. We further note that the scaling $s$ in the anterior–posterior direction is required to correct deviations from the slice thickness specified by the user, which in practice is never completely exact.

The optimal set of transforms is obtained by maximizing the objective function $F$ in *Equation 1*. This objective encodes four desired attributes of the reconstructed data, with relative weights α, β, γ, and υ :

1. The α term encourages a high overlap between the stack of $N$ 3D reconstructed slice masks $M\left[x;\{\Phi_n\},s\right]$ and the aligned reference volume $R\left[x;\Psi\right]$; we note that images are a function of spatial location $x$.
2. The β term promotes a high similarity between the image intensities of successive (reconstructed) slices $S_n\left[x;\Phi_n,s\right]$ and $S_{n+1}\left[x;\Phi_{n+1},s\right]$, for $n=1,\dots,N-1$.
3. The γ term encourages a high overlap between successive (reconstructed) slice masks $M_n\left[x;\Phi_n,s\right]$ and $M_{n+1}\left[x;\Phi_{n+1},s\right]$, for $n=1,\dots,N-1$.
4. The υ term promotes minimal scaling and shear in the 2D affine transforms; the function $f$ is a regularizer that prevents excessive deformation of the slices – particularly those showing little tissue, for example, the first and last slices of the stack.

$$
\begin{aligned}
F\left(\Psi,\{\Phi_n\}\right) = {} & \alpha D\left(M\left[x;\{\Phi_n\},s\right],R\left[x;\Psi\right]\right) + \beta\frac{1}{N-1}\sum_{n=1}^{N-1}C\left(S_n\left[x;\Phi_n,s\right],S_{n+1}\left[x;\Phi_{n+1},s\right]\right) \\
& +\gamma\frac{1}{N-1}\sum_{n=1}^{N-1}D\left(M_n\left[x;\Phi_n,s\right],M_{n+1}\left[x;\Phi_{n+1},s\right]\right) - \nu\frac{1}{N}\sum_{n=1}^{N}f\left(\Phi_n\right)
\end{aligned}
\tag{1}
$$

Mathematically, overlap in *Equation 1* is measured with the soft Dice coefficient $D$ (*Dice, 1945*; *Sorensen, 1948*; *Milletari et al., 2016*); image similarity is measured with the normalized cross-correlation $C$; and the scaling and shearing are measured with the absolute value of the logarithm of the determinant of the affine matrices, that is, $f\left(\Phi_n\right)$ is the absolute log-determinant of the $3\times 3$ matrix encoding $\Phi_n$. The relative weights $\alpha,\beta,\gamma,\nu$ are set via visual inspection of the output on a small pilot dataset. For the surface reference, we used: $\alpha=.95,\beta=\gamma=.025,\nu=\gamma/100$. For the probabilistic atlas, we trust the reference less and also use more regularization to prevent excessive deformation of slices: $\alpha=.8,\beta=\gamma=\nu=.1$. Either way, the 3D reconstruction is in practice not very sensitive to the exact value of these parameters. We also note that, as opposed to the preprocessing described in the previous section, SIFT is not a good candidate for matching consecutive slices: while it is resilient against changes in pose (e.g., object rotation), perspective, and lightning, it is not robust against changes in the object itself – such as changes between one slice to the next.

The objective function $F$ is minimized with standard numerical methods – specifically the limited-memory Broyden–Fletcher–Goldfarb–Shanno (LBFGS) algorithm (*Fletcher, 1987*). The LBFGS optimizer is initialized by stacking the photographs with their centers of gravity on coordinate (0,0), and then matching the center of gravity of the whole stack with the center of gravity of the 3D reference (as illustrated in *Video 1* in the supplement, see also https://youtu.be/wo5meYRaGUY).

If a probabilistic atlas is used instead of the surface scan, the same objective function is used but with a slightly different search space. Since we can no longer trust the external reference to correct for fine shape correction: (1) we keep the scaling factor of the anterior–posterior direction fixed to $s = 1$ (i.e., we trust the slice thickness specified by the user); (2) we use rigid rather than affine transforms $\{\phi_n\}$ for the slices (which also has the effect of making the regularizer equal to zero); and (3) we use a full 3D affine transform $\Psi$ for the reference. Sample reconstructions of a case with a 3D surface and a probabilistic atlas are shown in *Figure 5e–g*. The probabilistic atlas produces a plausible reconstruction, which is however far from the real shape of the specimen given by the surface (*Figure 5g*). An additional example from the MADRC dataset is shown in *Appendix 1—figure 2*. We note that 3D reconstruction is implemented in PyTorch, and runs efficiently on a GPU – less than 5 min on a Nvidia Quadro RTX 6000.

## Segmentation

The 3D reconstructed photographs have various applications (e.g., volumetry, computationally guided dissection) that require image segmentation, that is, assigning neuroanatomical labels to every spatial location (*Pham et al., 2000*; *Despotović et al., 2015*; *Akkus et al., 2017*). There are two main challenges when segmenting the 3D reconstructed photographs. First, the appearance of the images varies widely across cases due to differences in camera hardware (sensor, lens), camera settings, illumination conditions, and tissue preparation (e.g., fixed vs fresh). And second, accurate volumetry requires estimating the segmentation not only on the planes of the photographs but also in between slices.

In our previous work (*Tregidgo et al., 2020*), we adopted a Bayesian segmentation strategy that handled the first issue but not the second. Here, we extend a machine learning approach based on domain randomization (*Tobin et al., 2017*) that we have successfully applied to segment clinical brain MRI scans with large slice spacing (*Billot et al., 2020*; *Billot et al., 2023a*; *Billot et al., 2023b*). Specifically, our newly proposed approach 'Photo-SynthSeg' trains a convolutional neural network for image segmentation (a 3D U-Net, *Ronneberger et al., 2015*; *Çiçek et al., 2019*) with synthetic data as follows.

'Photo-SynthSeg' starts from a training dataset (the T1-39 dataset) that comprises a pool of 3D segmentations of brain images at isotropic 3D resolution (*Figure 6a*). At every iteration during training, one of these 3D segmentations is randomly selected and geometrically deformed with a random nonlinear transform, which simulates imperfect 3D reconstruction by including a 'deformation jitter' in the coronal direction, that is, small but abrupt deformations from one coronal slice to the next (*Figure 6b*). This deformed isotropic segmentation is used to generate a synthetic 3D image (also at isotropic resolution) using a Gaussian mixture model conditioned on the segmentation (*Figure 6c*). Crucially, we randomize the Gaussian parameters (means and variances) to make the neural network

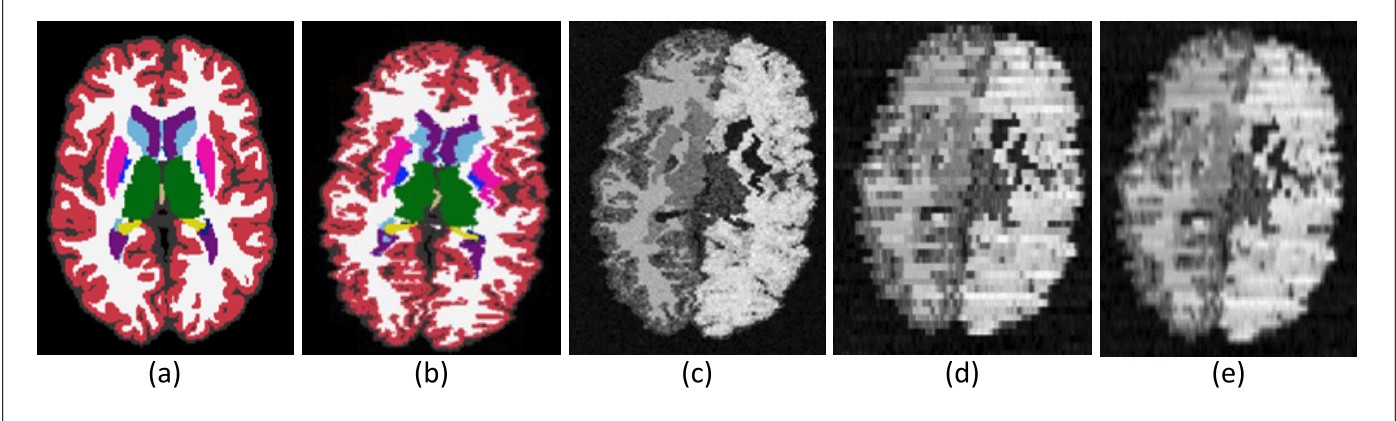

(a)        (b)        (c)        (d)        (e)

**Figure 6.** Intermediate steps in the generative process. (**a**) Randomly sampled input label map from the training set. (**b**) Spatially augmented input label map; imperfect 3D reconstruction is simulated with a deformation jitter across the coronal plane. (**c**) Synthetic image obtained by sampling from a Gaussian mixture model conditioned on the segmentation, with randomized means and variances. (**d**) Slice spacing is simulated by downsampling to low resolution. This imaging volume is further augmented with a bias field and intensity transforms (brightness, contrast, gamma). (**e**) The final training image is obtained by resampling (**d**) to high resolution. The neural network is trained with pairs of images like (**e**) (input) and (**b**) (target).

robust against variations in image appearance. This synthetic isotropic image is then 'digitally sliced' into a 3D stack of synthetic coronal photographs, each of which is independently corrupted by a random 2D smooth illumination field that simulates inhomogeneities in the illumination of the photographs (*Figure 6d*). Importantly, we also randomize the thickness of the simulated slices in every iteration and introduce small stochastic variations in thickness around the central value within each simulation. This strategy makes the network agnostic to the slice thickness of the reconstructed 3D stack at test time. Finally, the simulated stack, which has anisotropic resolution (e.g., 1 mm in-plane and several mm in slice thickness), is resampled to the original isotropic resolution of the (deformed) 3D segmentation (*Figure 6e*) – typically 1 mm isotropic.

This generative process mimics the image formation process in the real world and has two crucial aspects. First, the super-resolution component: Photo-SynthSeg is a U-Net that will produce a high-resolution (1 mm) isotropic 3D segmentation for every input at test time, independently of the slice spacing of the 3D reconstructed stack of photographs (*Figure 2*). And second, domain randomization: sampling different Gaussian parameters, illumination fields, and slice thicknesses at every iteration beyond realistic limits (as in *Figure 6*, where even contralateral regions have different appearance) forces the U-Net to learn features that are independent of the intensity profiles of the photographs, as well as of the spacing between slices. This process makes the U-Net robust against changes in the acquisition. We note that the Gaussian distributions are univariate rather than trivariate, that is, they model grayscale rather than red–green–blue triplets; we tried training a U-Net with the latter, but the performance on color images was worse than when converting them to grayscale and using the U-Net trained with univariate Gaussians.

During training, resampled stacks and corresponding segmentations (*Figure 6b–e*) are fed to a U-Net. The U-Net architecture is the same as in our previous works with synthetic scans (*Billot et al., 2020*; *Billot et al., 2023a*): it consists of five levels, each separated by a batch normalization layer (*Ioffe and Szegedy, 2015*) along with a max-pooling (contracting path) or an upsampling operation (expanding path). All levels comprise two convolution layers with $3 \times 3 \times 3$ kernels. Every convolutional layer is associated with an Exponential Linear Unit activation (*Clevert et al., 2016*), except for the last one, which uses a softmax. While the first layer comprises 24 feature maps, this number is doubled after each max-pooling, and halved after each upsampling. Following the original U-Net architecture, skip connections are used across the contracting and expanding paths. The network is trained with a soft Dice loss (*Milletari et al., 2016*) and the Adam optimizer (*Kingma and Ba, 2014*). The deep learning model is implemented in Keras (*Chollet, 2015*) with a Tensorflow backend (*Abadi et al., 2016*) and runs on a few seconds on a Nvidia Quadro RTX 6000 GPU. Training takes around 7 days on the same GPU.

Finally, we note that there are two different versions of Photo-SynthSeg: one for full cerebra and one for single hemispheres. The later is trained with left hemispheres and flipped right hemispheres, and can thus be used as is for left hemispheres. To process a right hemisphere, we simply left–right flip it, segment it, and flip the results back. While the default Photo-SynthSeg pipeline segments the cerebral cortex as a whole (i.e., as in *Figure 6a*), our tool also offers the option of subdividing it into parcels as defined by the Desikan–Killiany atlas (*Desikan et al., 2009*), like in *Figure 2*. This is achieved with the cortical parcellation module (Segmenter S3) of our tool 'SynthSeg' (*Billot et al., 2023a*), which is also distributed with FreeSurfer.

## Acknowledgements

First and foremost, we thank the research participants who donated their brains to science and their families, without whom this work would be impossible. This research was supported by primarily supported by the National Institute of Aging (R01AG070988 and P30AG062421). Other support was provided by NIH grants RF1MH123195, R01EB031114, UM1MH130981, P30AG066509 (UW ADRC), U19AG066567, U19AG060909, K08AG065426, and R01NS112161; the European Union (ERC Starting Grant 677697); Alzheimer's Research UK (ARUK-IRG2019A-003); and the Massachusetts Life Sciences Center. AC was further funded by the European Union, the Ministry of Universities and Recovery, and the Transformation and Resilience Plan, through a call from Universitat Politècnica de Catalunya (Grant Ref 2021UPC-MS-67573). Additional support was provided by NIH grants U01MH117023, 1R01EB023281, R01EB006758, R21EB018907, R01EB019956, P41EB030006, 1R56AG064027, 1R01AG064027, 5R01AG008122, R01AG016495, 1R01AG070988, UM1MH130981, R01 MH123195,

R01 MH121885, 1RF1MH123195, R01NS0525851, R21NS072652, R01NS070963, R01NS083534, 5U01NS086625,5U24NS10059103, R01NS105820, 1S10RR023401, 1S10RR019307, 1S10RR023043, and 5U01MH093765, and by the Buster Alvord Endowment. BF has a financial interest in Cortico-Metrics, a company developing brain MRI measurement technology. His interests are reviewed and managed by Massachusetts General Hospital.

## Additional information

### Competing interests

Bruce Fischl: BF has a financial interest in CorticoMetrics, a company developing brain MRI measure-menttechnology; his interests are reviewed and managed by Massachusetts General Hospital. Bradley T Hyman: Reviewing editor, *eLife*. The other authors declare that no competing interests exist.

### Funding

| Funder | Grant reference number | Author |
|---|---|---|
| National Institute on Aging | R01AG070988 | Juan E Iglesias |
| National Institute on Aging | P30AG062421 | Bradley T Hyman |
| National Institutes of Health | RF1MH123195 | Juan E Iglesias |
| National Institutes of Health | R01EB031114 | Juan E Iglesias |
| National Institutes of Health | UM1MH130981 | Juan E Iglesias |
| National Institutes of Health | P30AG066509 (UW ADRC) | C Dirk Keene |
| National Institutes of Health | U19AG066567 | C Dirk Keene |
| National Institutes of Health | U19AG060909 | C Dirk Keene |
| National Institutes of Health | K08AG065426 | Caitlin S Latimer |
| National Institutes of Health | R01NS112161 | Koen Van Leemput |
| European Union | ERC Starting Grant 677697 | Juan E Iglesias |
| Alzheimer's Research UK | ARUK-IRG2019A-003 | Juan E Iglesias |
| Politècnica de Catalunya | Grant Ref 2021UPC-MS-67573 | Adria Casamitjana |
| National Institutes of Health | U01MH117023 | Bruce Fischl |
| National Institutes of Health | 1R01EB023281 | Bruce Fischl |
| National Institutes of Health | R01EB006758 | Bruce Fischl |
| National Institutes of Health | R21EB018907 | Bruce Fischl |
| National Institutes of Health | R01EB019956 | Bruce Fischl |
| National Institutes of Health | P41EB030006 | Bruce Fischl |

| Funder | Grant reference number | Author |
|---|---|---|
| National Institutes of Health | 1R56AG064027 | Bruce Fischl |
| National Institutes of Health | 1R01AG064027 | Bruce Fischl |
| National Institutes of Health | 5R01AG008122 | Bruce Fischl |
| National Institutes of Health | R01AG016495 | Bruce Fischl |
| National Institutes of Health | 1R01AG070988 | Juan E Iglesias |
| National Institutes of Health | UM1MH130981 | Juan E Iglesias |
| National Institutes of Health | R01 MH123195 | Juan E Iglesias |
| National Institutes of Health | R01 MH121885 | Bruce Fischl |
| National Institutes of Health | 1RF1MH123195 | Juan E Iglesias |
| National Institutes of Health | R01NS0525851 | Bruce Fischl |
| National Institutes of Health | R21NS072652 | Bruce Fischl |
| National Institutes of Health | R01NS070963 | Bruce Fischl |
| National Institutes of Health | R01NS083534 | Bruce Fischl |
| National Institutes of Health | 5U01NS086625 | Bruce Fischl |
| National Institutes of Health | 5U24NS10059103 | Bruce Fischl |
| National Institutes of Health | R01NS105820 | Bruce Fischl |
| National Institutes of Health | 1S10RR023401 | Bruce Fischl |
| National Institutes of Health | 1S10RR019307 | Bruce Fischl |
| National Institutes of Health | 1S10RR023043 | Bruce Fischl |
| National Institutes of Health | 5U01MH093765 | Bruce Fischl |

The funders had no role in study design, data collection, and interpretation, or the decision to submit the work for publication.

## Author contributions

Harshvardhan Gazula, Conceptualization, Software, Validation, Investigation, Methodology, Writing - original draft, Writing – review and editing; Henry FJ Tregidgo, Adria Casamitjana, Conceptualization, Software, Investigation, Methodology, Writing - original draft, Writing – review and editing; Benjamin Billot, Software, Investigation, Methodology, Writing – review and editing; Yael Balbastre, Software, Methodology, Writing – review and editing; Jonathan Williams-Ramirez, Rogeny Herisse, Lucas J Deden-Binder, Resources, Data curation, Investigation; Erica J Melief, Caitlin S Latimer, Conceptualization, Resources, Writing – review and editing; Mitchell D Kilgore, Michael S Marshall, Resources, Data curation; Mark Montine, Resources, Data curation, Writing – review and editing;

Eleanor Robinson, Theresa R Connors, Data curation, Investigation, Writing – review and editing; Emily Blackburn, Resources, Investigation, Writing – review and editing; Derek H Oakley, Conceptualization, Investigation, Writing – review and editing; Matthew P Frosch, Christine L MacDonald, C Dirk Keene, Conceptualization, Funding acquisition, Writing – review and editing; Sean I Young, Investigation, Methodology, Writing – review and editing; Koen Van Leemput, Conceptualization, Methodology, Writing – review and editing; Adrian V Dalca, Bruce Fischl, Conceptualization, Writing – review and editing; Bradley T Hyman, Conceptualization, Investigation, Methodology, Writing – review and editing; Juan E Iglesias, Conceptualization, Software, Funding acquisition, Investigation, Visualization, Writing - original draft, Writing – review and editing

### Author ORCIDs
Henry FJ Tregidgo (ID) http://orcid.org/0000-0002-3509-8154
Mitchell D Kilgore (ID) http://orcid.org/0000-0003-1101-6924
C Dirk Keene (ID) http://orcid.org/0000-0002-5291-1469
Bradley T Hyman (ID) http://orcid.org/0000-0002-7959-9401
Juan E Iglesias (ID) http://orcid.org/0000-0001-7569-173X

### Ethics

MADRC data: The Institutional Review Board of Massachusetts General Hospital approved the study, and consent was obtained from the patient's authorized representative at the time of death. UW-ADRC data: In 2016, Institutional Review Board of the University of Washington (UW) issued an official determination that our repository work does not meet the metric of human subject's research as we are working solely with deceased individuals. Our practices are now informed by the US Revised Uniform Anatomical Gift ACT 2006 (Last Revised or Amended in 2009) and Washington Statute Chapter 68.64 RCW. Furthermore, we work closely with the UW School of Medicine Compliance Office on our consent forms and HIPAA compliance. All materials are indeed collected under informed consent.

Reviewer #1 (Public review): https://doi.org/10.7554/eLife.91398.4.sa1
Reviewer #2 (Public review): https://doi.org/10.7554/eLife.91398.4.sa2
Author response https://doi.org/10.7554/eLife.91398.4.sa3

## Additional files

### Supplementary files
• MDAR checklist

### Data availability

HCP is publicly available at https://www.humanconnectome.org/study/hcp-young-adult/data-releases. The MADRC and UW-ADRC datasets are available on Datalad. Instructions for retrieving them can be found at https://github.com/MGH-LEMoN/elife-data (copy archived at *Gazula, 2024*). We note that the T1-39 dataset cannot be released because it was collected under a set of Institutional Review Boards that did not include a provision approving public distribution. The code used for training segmentation models using the T1-39 dataset can be found at https://github.com/BBillot/SynthSeg (copy archived at *Billot, 2024*).

The following previously published dataset was used:

| Author(s) | Year | Dataset title | Dataset URL | Database and Identifier |
|---|---|---|---|---|
| Glasser MF | 2013 | Human Connectome Project | https://www.humanconnectome.org/study/hcp-young-adult/document/900-subjects-data-release | Humanconnectome, 900-subjects-data-release |

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

## Appendix 1

### Data acquisition at the Massachusetts Alzheimer's Disease Research Center (MADRC)

The whole brain is removed using standard autopsy procedures. After removal of the dura matter, the brain is weighed, photo-documented, and then fixed in 10% neutral-buffered formalin (NBF). Some specimens are cut along the midline sagittal plane with one hemibrain frozen for future biochemical studies. After at least 1 week of fixation, the specimens are photo-documented again. The posterior fossa is then dissected off the fixed full or hemibrain through the midbrain at the level of cranial nerve III, and the supratentorial portion of the brain is 3D surface scanned.

After scanning, the brain or hemibrain is sectioned at defined anatomical landmarks (anterior temporal tips, optic chiasm, infundibulum, mamillary bodies, cerebral peduncles, red nuclei, and colliculi) with the frontal and occipital lobes additionally sectioned at approximately 10 mm intervals. The front and back of all sections are then photographed with a metric ruler included for size reference. The photographs are then deidentified for further analysis.

### Data acquisition at the Alzheimer's Disease Research Center of the University of Washington (UW-ADRC)

The whole brain is removed using standard autopsy procedures. The brain is weighed and photo-documented, and then the meninges are removed. For donors with a *post mortem* interval (PMI: the time interval between death and procurement) longer than 12 hr, the whole brain is fixed in 10% NBF with no tissue frozen. A rapid slicing protocol is performed for donors with a *post mortem* interval of fewer than 12 hr. For a rapid protocol, the brain is bisected along the midline, and one hemibrain is placed in a bucket with 10% NBF for 2 weeks. The other hemibrain is 3D surface scanned, then placed in a scaffolding box with the vermis of the cerebellum flush against the posterior wall. The hemibrain is embedded in freshly mixed dental alginate (540 g powder blended in 4 l of water) and submerged until the alginate is set. The alginate block containing the hemibrain is placed in a slicing sled with the frontal pole toward the front. The hemibrain is sliced anterior to posterior in 4 mm slices. All slices are set on Teflon-coated aluminum plates and photographed. Alternating slices are set to be flash-frozen, with the remaining slices fixed in 10% NBF. The hemibrain (rapid) or whole (non-rapid) brain is fixed in 10% NBF for 2 weeks, embedded in agarose, and scanned in a 3T MRI scanner.

Following the MRI, the agarose-embedded brain is sliced on a modified deli slicer for precise 4 mm tissue slabs aligned with the ex vivo MRI images for image-guided tissue sampling to standard tissue sampling following the current National Institute of Aging-Alzheimer's Association (NIA-AA) consensus guidelines. After sampling, all blocks are processed and embedded in paraffin according to standard techniques. Following current NIA-AA guidelines, the resulting Fast-Frozen Paraffin-Embedded blocks are cut and stained for diagnostic analysis. Histologically stained slides are scanned into the HALO (https://indicalab.com/halo/) imaging workstation using an Aperio AT2 slide scanner and are analyzed by a trained neuropathologist.

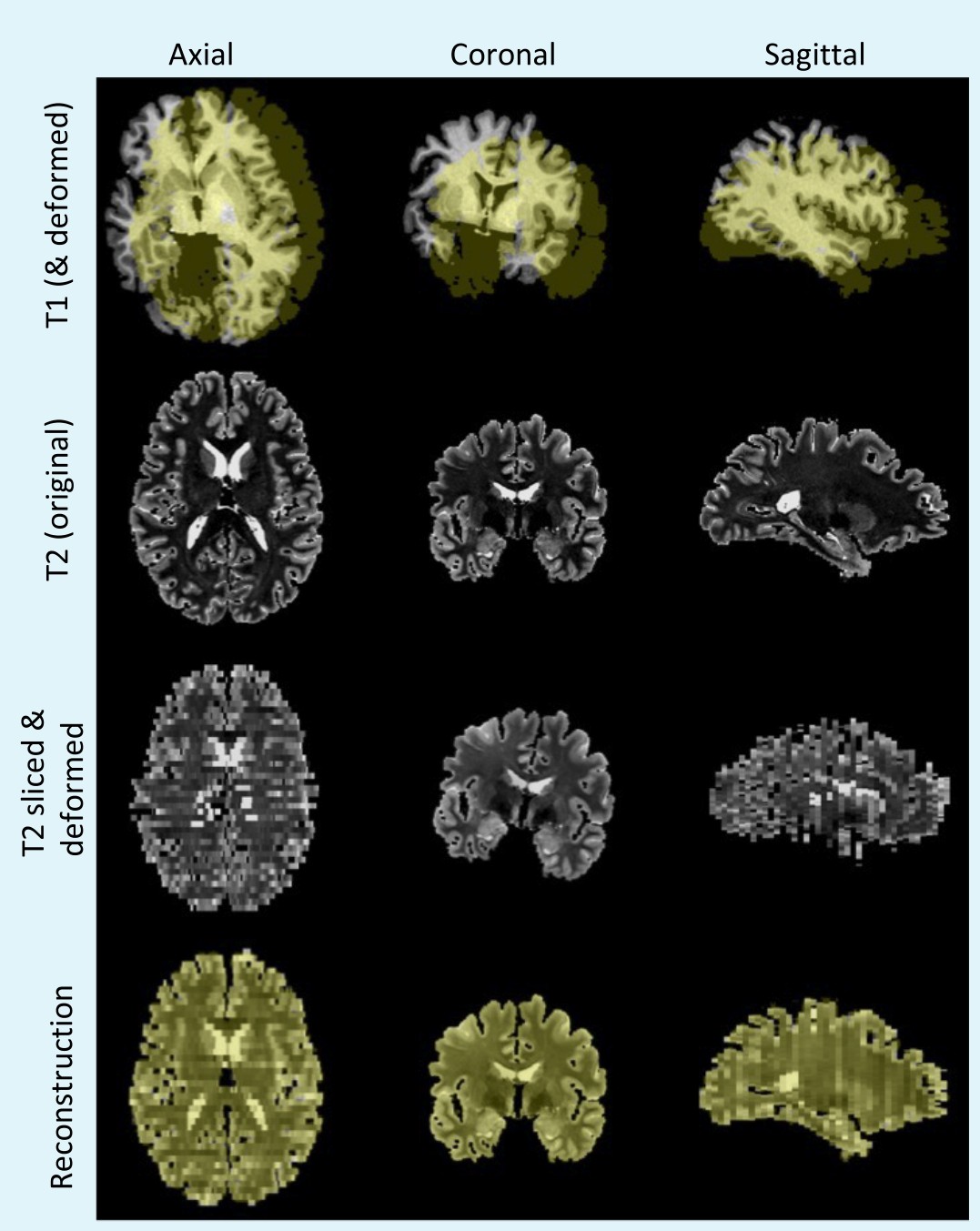

**Appendix 1—figure 1.** Simulation and reconstruction of synthetic data. Top row: skull stripped T1 scan and (randomly translated and rotated) binary mask of the cerebrum, in yellow. Second row: original T2 scan. Third row: randomly sliced and linearly deformed T2 images. Bottom row: output of the 3D reconstruction algorithm, that is, reconstructed T2 slices and registered reference mask overlaid in yellow.

(a) (b) (c)

**Appendix 1—figure 2.** Reconstruction with surface scan vs probabilistic atlas. (**a**) Initialization, with contour of 3D surface scan superimposed. (**b**) Reconstruction with 3D surface scan. (**c**) Reconstruction with probabilistic atlas (overlaid as heat map with transparency); the contour of the surface scan is overlaid in light blue, for comparison. Even though the shape of the reconstruction in (**c**) is plausible, it is clearly inaccurate in light of the surface scan.

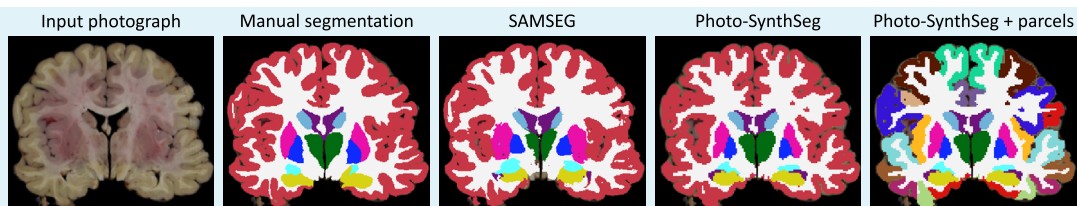

| Input photograph | Manual segmentation | SAMSEG | Photo-SynthSeg | Photo-SynthSeg + parcels |

**Appendix 1—figure 3.** Example of mid-coronal slice selected for manual segmentation and computation of Dice scores. Compared with the FreeSurFer protocol, we merge the ventral diencephalon (which has almost no visible contrast in the photographs) with the cerebral white matter in our manual delineations. We also merged this structures in the automated segmentations from SAMSEG and Photo-SynthSeg in this figure, for a more consistent comparison.

