## [Editor Report · eLife assessment]

The authors of this study implemented an **important** toolset for 3D reconstruction and segmentation of dissection photographs, which could serve as an alternative for cadaveric and ex vivo MRIs. The tools were tested on synthetic and real data with **compelling** performance. This toolset could further contribute to the study of neuroimaging-neuropathological correlations.

---

## [Referee Report · Reviewer #1 (Public review)]

Gazula and co-workers presented in this paper a software tool for 3D structural analysis of human brains, using slabs of fixed or fresh brains. This tool will be included in Freesurfer, a well-known neuroimaging processing software. It is possible to reconstruct a 3D surface from photographs of coronal sliced brains, optionally using a surface scan as model. A high-resolution segmentation of 11 brain regions is produced, independent of the thickness of the slices, interpolating information when needed. Using this method, the researcher can use the sliced brain to segment all regions, without the need of ex vivo MRI scanning.

The software suite is freely available and includes 3 modules. The first accomplishes preprocessing steps, for correction of pixel sizes and perspective. The second module is a registration algorithm that registers a 3D surface scan obtained prior to sectioning (reference) to the multiple 2D slices. It is not mandatory to scan the surface, -a probabilistic atlas can also be used as reference- however the accuracy is lower. The third module uses machine learning to perform the segmentation of 11 brain structures in the 3D reconstructed volume. This module is robust, dealing with different illumination conditions, cameras, lens and camera settings. This algorithm ("Photo-SynthSeg") produces isotropic smooth reconstructions, even in high anisotropic datasets (when the in-plane resolution of the photograph is much higher than the thickness), interpolating the information between slices.

To verify the accuracy and reliability of the toolbox, the authors reconstructed 3 datasets, using real and synthetic data. Real data of 21 postmortem confirmed Alzheimer's disease cases from the Massachusetts Alzheimer's Disease Research Center (MADRC)and 24 cases from the AD Research at the University of Washington(who were MRI scanned prior to processing)were employed for testing. These cases represent a challenging real-world scenario. Additionally, 500 subjects of the Human Connectome project were used for testing error as a continuous function of slice thickness. The segmentations were performed with the proposed deep-learning new algorithm ("Photo-SynthSeg") and compared against MRI segmentations performed to "SAMSEG" (an MRI segmentation algorithm, computing Dice scores for the segmentations. The methods are sound and statistically showed correlations above 0.8, which is good enough to allow volumetric analysis. The main strengths of the methods are the datasets used (real-world challenging and synthetic) and the statistical treatment, which showed that the pipeline is robust and can facilitate volumetric analysis derived from brain sections and conclude which factors can influence in the accuracy of the method (such as using or not 3D scan and using constant thickness)).

Although very robust and capable of handling several situations, the researcher has to keep in mind that processing has to follow some basic rules in order for this pipeline to work properly. For instance, fiducials and scales need to be included in the photograph, and the slabs should be photographed against a contrasting background. Also, only coronal slices can be used, which can be limiting for certain situations.

The authors achieved their aims, and the statistical analysis confirms that the machine learning algorithm performs segmentations comparable to the state-of-the-art of automated MRI segmentations.

Those methods will be particularly interesting to researchers who deal with post-mortem tissue analysis and do not have access to ex vivo MRI. Quantitative measurements of specific brain areas can be performed in different pathologies and even in the normal aging process. The method is highly reproducible, and cost-effective since allows the pipeline to be applied by any researcher with small pre-processing steps.

---

## [Referee Report · Reviewer #2 (Public review)]

Summary

The authors proposed a toolset Photo-SynthSeg to the software FreeSurfer which performs 3D reconstruction and high-resolution 3D segmentation on a stack of coronal dissection photographs of brain tissues. To prove the performance of the toolset, three experiments were conducted, including volumetric comparison of brain tissues on AD and HC groups from MADRC, quantitative evaluation of segmentation on UW-ADRC and quantitative evaluation of 3D reconstruction on HCP digitally sliced MRI data.

Strengths

To guarantee the successful workflow of the toolset, the authors clearly mentioned the prerequisites of dissection photograph acquisition, such as fiducials or rulers in the photos and tissue placement of brain slices with more than one connected component. The quantitative evaluation of segmentation and reconstruction on synthetic and real data demonstrates the accuracy of the methodology. Also, the successful application of this toolset on two brain banks with different slice thicknesses, tissue processing and photograph settings demonstrates its robustness. By working with tools of the SynthSeg pipeline, Photo-SynthSeg could further support volumetric cortex parcellation. The toolset also benefits from its adaptability of different 3D references, such as surface scan, ex vivo MRI and even probabilistic atlas, suiting the needs for different brain banks.

Weaknesses

Certain weaknesses are already covered in the manuscript. Cortical tissue segmentation could be further improved. The quantitative evaluation of 3D reconstruction is quite optimistic due to random affine transformations. Manual edits of slice segmentation task are still required and take a couple of minutes per photograph. Finally, the current toolset only accepts coronal brain slices and should adapt to axial or sagittal slices in future work.

---

## [Author Response]

The following is the authors’ response to the previous reviews.

Is the coronal slice in Figure 2 the corresponding mid-coronal plane to compute Dice scores? If so, the authors could mention it so that readers have an idea where the selected slice is.

This is indeed a good point. The coronal slice in Figure 2 is not part of the set of slices that we used to compute Dice scores. Showing such a slice is important, so we have added a small figure to the appendix with one of these slices, along with the corresponding automated segmentations.

SIFT descriptors were adopted to detect fiducials only. Maybe it could also be applied to align stacked photographs of brain slices.

While SIFT is robust against changes in pose (e.g., object rotation), perspective, and lightning, it is not robust against changes in the object itself – such as changes between one slice to the next, as is the case in our work. We have added a sentence to the methods section clarifying this issue.